# Bayesian Network Structure Learning using Digital Annealer

## Abstract

Annealing processors, which efficiently solve a quadratic unconstrained binary optimization (QUBO), are a potential breakthrough in improving the accuracy of score-based Bayesian network structure learning. However, currently, the bit capacity of an annealing processor is very limited. To utilize the power of annealing processors, it is necessary to encode score-based learning problems into QUBO within the upper bound of bits. In this paper, we propose a novel approach with direct encoding of candidate parent sets in the form of Cartesian products. Experimental results on benchmark networks with 27 to 70 variables show that our approach requires lesser bits than the bit capacity of the second-generation Fujitsu digital annealer, a fully coupled annealing processor developed by with semiconductor technology. Moreover, we demonstrate that the digital annealer with our conversion method consistently outperforms the state-of-the-art heuristic algorithms on the benchmark networks.

## 1 Introduction

A Bayesian network is a probabilistic graphical model that represents the structure of a joint probability distribution among random variables in a directed acyclic graph (DAG) [Pearl, 1988]. One class of associated computational problems is learning the structure of a Bayesian network from data. We focus on score-based Bayesian network structure learning for finding the DAG with a maximal score that depends on the data [Cooper and Herskovits, 1992, Cowell, 2001].

The Bayesian network learning problem is NP-hard [Chickering et al., 2004]; therefore, the standard methodology is using heuristic approaches. Many algorithms have been proposed to improve the accuracy and to reduce the running time. A search over the space of orderings [Teyssier and Koller, 2005, Scanagatta et al., 2015] is one of the most successful heuristic approaches.

Annealing processors may contribute to finding a high-scoring network structure in a realistic timeframe. An annealing processor is expected to be an alternative hardware to von Neumann computers for quadratic unconstrained binary optimization (QUBO) problems. In particular, it is reported that complementary metal oxide semiconductor (CMOS) annealing processors already outperform conventional computers on the speed of solving max-cut problems [Gyoten et al., 2018].

We note that the bit capacity of an annealing processor is currently limited. Therefore, we need an efficient conversion method of Bayesian network structure learning into QUBO within the limited bits. Additionally, it is also important to show the lower bounds of penalty coefficients because the precision for the biases and variable couplers is limited.

Submitted to 35th Conference on Neural Information Processing Systems (NeurIPS 2021). Do not distribute.

Annealing processors are classified into the nearest neighbor type and the fully connected type [Yamamoto, 2020]. While the coupling nodes of a nearest neighbor annealing processor is limited to only between adjacent nodes, the coupling exists between arbitrary nodes of a fully coupled annealing processor. Though the scalability of nearest neighbor annealing processors is high, it is necessary to consider the additional bits for minor embedding [Choi, 2008, 2010].

O'Gorman et al. 2014 proposed a method to convert score-based Bayesian network structure learning into QUBO that requires $\mathcal{O}(n^2)$ bits for $n$ random variables and a maximum parent set size $m = 2$. They also demonstrated the sufficient lower bounds of penalty coefficients. However, when $m \geq 3$, the number of necessary auxiliary variables for a quadratization [Boros and Gruber, 2014] is at most $\mathcal{O}(n(n-1)^{\frac{m}{2}})$. This is a significant disadvantage for the current limited bit capacity of annealing processors.

In this study, we propose an efficient conversion method based on the advanced identification of candidate parent sets and their representation in the form of Cartesian products. We also provide a greedy algorithm to decompose the candidate parent sets into the form of Cartesian products and prove the sufficient lower bounds of penalty coefficients.

Experimental results on benchmark networks with 27 to 70 variables show that our conversion method reduces the required bits significantly in comparison to the previous work [O'Gorman et al., 2014]. Our approach allows us to utilize the power of the second generation Fujitsu digital annealer, a fully coupled CMOS annealing processor [Aramon et al., 2019]. We demonstrate that the digital annealer consistently outperforms the ordering space search algorithms on the benchmark networks.

## 2 Background

### 2.1 Score-based Bayesian Network Structure Learning

The goal of score-based Bayesian network structure learning is to find a DAG with maximal score. Given to random variables $\mathcal{X} = (X_i)_{i=1}^n$ and a complete data set of $N$ instances $\mathcal{D} = \{D_1, \cdots, D_N\}$, we optimize the parent set $\Pi_i$ of each random variable,

$$\Pi_1^*, \cdots, \Pi_n^* = \operatorname*{arg\,min}_{\substack{\Pi_1, \cdots, \Pi_n \subset \mathcal{X} \\ \mathcal{G} \in \text{DAG}}} \sum_{i=1}^n -\log S^{(i)}(\Pi_i \mid \mathcal{D}), \tag{1}$$

where $\mathcal{G} = (\mathcal{V}, \mathcal{E}), \mathcal{V} = \{1, \cdots, n\}, \mathcal{E} = \{(j, i) \mid j, i \in \{1, \cdots, n\}, X_j \in \Pi_i\}$, and $S_i : \Pi_i \to \mathbb{R}$ is a local score function corresponding to $X_i$. The Bayesian Dirichlet equivalent uniform (BDeu) score [Buntine, 1991] is one of the commonly used scores,

$$S_{\text{BDeu}}^{(i)}(\Pi_i \mid \mathcal{D}) \equiv \prod_{j=1}^{\beta_i} \frac{\Gamma(\alpha_{i,j})}{\Gamma(N_{i,j} + \alpha_{i,j})} \prod_{k=1}^{\gamma_i} \frac{\Gamma(N_{i,j,k} + \alpha_{i,j,k})}{\Gamma(\alpha_{i,j,k})}, \tag{2}$$

where $N = \sum_{j=1}^{\beta_i} N_{i,j}, N_{i,j} = \sum_{k=1}^{\gamma_i} N_{i,j,k}, \alpha_{i,j} = \sum_{k=1}^{\gamma_i} \alpha_{i,j,k}$, $\beta_i$ is the number of joint states of $\Pi_i$, $\gamma_i$ is the number of states of $X_i$, $N_{i,j,k}$ is the number of cases of the parent set $\Pi_i$ in its $j$-th state and $X_i$ in its $k$-th state, $\alpha_{i,j,k} = \frac{\alpha}{\beta_i \gamma_i}$ is the hyperparameter of the Dirichlet function, and $0 < \alpha \in \mathbb{R}$ is called equivalent sample size [Heckerman et al., 1995a].

### 2.2 Hamiltonian

The Hamiltonian, which is the objective function of an annealing processor, is a quadratic pseudo-Boolean function,

$$H(\boldsymbol{\sigma}) = \sum_{i \in \mathcal{V}_{\text{AP}}} h_i \sigma_i + \sum_{(i,j) \in \mathcal{E}_{\text{AP}}} J_{i,j} \sigma_i \sigma_j, \tag{3}$$

where $\boldsymbol{\sigma} = (\sigma_i)_{i=1}^{|\mathcal{V}_{\text{AP}}|} \in \mathbb{B}^{|\mathcal{V}_{\text{AP}}|}$, the biases $h_i \in \mathbb{R}$ for all $i \in \mathcal{V}_{\text{AP}}$, the couplers $J_{i,j} \in \mathbb{R}$ for all $(i, j) \in \mathcal{E}_{\text{AP}}$, and the graph $\mathcal{G}_{\text{AP}} = (\mathcal{V}_{\text{AP}}, \mathcal{E}_{\text{AP}})$. Higher degree problems are reformed into quadratic ones using auxiliary variables. This reformulation is called quadratization.

**Definition 1.** If a quadratic polynomial function $g(\boldsymbol{v}, \boldsymbol{h})$ is a quadratization of a pseudo-Boolean function $f(\boldsymbol{v})$, then $f(\boldsymbol{v}) = \min_{\boldsymbol{h} \in \mathbb{B}^J} g(\boldsymbol{v}, \boldsymbol{h})$ for all $\boldsymbol{v} \in \mathbb{B}^I$.

Anthony et al. 2016 proved that every pseudo-Boolean function of $I$ variables and of degree $K$ has a quadratization involving at most $\mathcal{O}(I^{\frac{K}{2}})$ auxiliary variables. In particular, at most $\mathcal{O}(2^{\frac{I}{2}})$ when $K = I$. It is well known that every pseudo-Boolean function can be uniquely represented as a multilinear polynomial in its variables [Boros and Hamme, 2002].

## 2.3 Basic Conversion of Score-based Bayesian Network Structure Learning

Using $n(n-1)$ bits to encode the paths into $\boldsymbol{d} = ((d_{j,i})_{1 \leq j \leq n, j \neq i})_{i=1}^n \in \mathbb{B}^{n(n-1)}$ ($d_{j,i} = 1$ if $X_j$ is the parent of $X_i$, $d_{j,i} = 0$ otherwise) and $\binom{n}{2}$ bits to encode the topological orders into $\boldsymbol{r} = (r_{i,j})_{1 \leq i < j \leq n} \in \mathbb{B}^{\binom{n}{2}}$ ($r_{i,j} = 0$ if the order of $X_i$ is higher than $X_j$, $r_{i,j} = 1$ otherwise), it is possible to represent eq. (1) on the Hamiltonian,

$$H_{\text{total}}(\boldsymbol{d}, \boldsymbol{r}) \equiv \sum_{i=1}^n H_{\text{score}}^{(i)}(\boldsymbol{d}_{\cdot,i}) + H_{\text{cycle}}(\boldsymbol{d}, \boldsymbol{r}). \tag{4}$$

The states of $\boldsymbol{d}_{\cdot,i}$ are mapped one-to-one to the states of $\Pi_i$. Let $\Pi_i = \pi^{(i)}(\boldsymbol{d}_{\cdot,i})$ for all $1 \leq i \leq n$. The local score of the Hamiltonian is

$$H_{\text{score}}^{(i)}(\boldsymbol{d}_{\cdot,i}) \equiv -\log S^{(i)}(\pi^{(i)}(\boldsymbol{d}_{\cdot,i}) \mid \mathcal{D}) + \log S^{(i)}(\phi \mid \mathcal{D}), \tag{5}$$

for all $1 \leq i \leq n$. The score function has a quadratization involving at most $\mathcal{O}(n2^{\frac{n-1}{2}})$ auxiliary variables. O'Gorman et al. 2014 added the maximum parent set size constraint to the Hamiltonian. In this case, the number of auxiliary variables is at most $\mathcal{O}(n(n-1)^{\frac{m}{2}})$. The cycle constraint of the Hamiltonian consists of the topological order constraint and the consistency constraint,

$$H_{\text{cycle}}(\boldsymbol{d}, \boldsymbol{r}) \equiv \sum_{1 \leq i < j < k \leq n} \delta_1 R(r_{i,j}, r_{j,k}, r_{i,k}) + \sum_{1 \leq i < j \leq n} \delta_2 (d_{i,j} r_{i,j} + d_{j,i}(1 - r_{i,j})), \tag{6}$$

where $R(r_1, r_2, r_3) = r_1 r_2 (1 - r_3) + (1 - r_1)(1 - r_2) r_3$ for all $r_1, r_2, r_3 \in \mathbb{B}$. When the penalty coefficients $0 < \delta_1, \delta_2 \in \mathbb{R}$ are sufficiently large, the DAG constraint is satisfied indirectly through the relationship of the paths $\boldsymbol{d}$ and the topological order $\boldsymbol{r}$. If it holds that

$$\max\{0, \max_{\substack{1 \leq j^*, i^* \leq n \\ j^* \neq i^*}} \max_{\substack{\boldsymbol{d}_{\cdot,i^*} \in \mathbb{B}^{n-1} \\ d_{j^*,i^*}=1}} (H_{\text{score}}^{(i^*)}(\boldsymbol{d}_{\cdot,i^*}^{(j^*,i^*)}) - H_{\text{score}}^{(i^*)}(\boldsymbol{d}_{\cdot,i^*}))\} < \delta_1 < \frac{\delta_2}{n-2}, \tag{7}$$

then there is no cycle on the paths of the ground state, where $\boldsymbol{d}^{(j^*,i^*)} = ((d_{j,i})_{1 \leq j \leq n, j \neq i}^{(j^*,i^*)})_{i=1}^n$, $d_{j,i}^{(j^*,i^*)} = 0$ if $(j,i) = (j^*, i^*)$, $d_{j,i}^{(j^*,i^*)} = d_{j,i}$ otherwise. The computational cost to obtain the left side of eq. (7) is at most $\mathcal{O}(n^{m+1})$. In particular, at most $\mathcal{O}(n^2 2^{n-2})$ when $m = n - 1$.

# 3 Candidate Parent Set Decomposition

Parent set identification is a major technique to narrow the search space of structure optimization, based on the relationship between parent sets and local scores under the DAG constraints [de Campos and Ji, 2011, Correia et al., 2020]. The collection of candidate parent sets of a random variable $X_i$ $\{W \subseteq \mathcal{X} \setminus \{X_i\} \mid W' \subset W \Rightarrow S^{(i)}(W' \mid \mathcal{D}) < S^{(i)}(W \mid \mathcal{D})\}$. To reduce the required bits of the score component of the Hamiltonian, we propose an efficient conversion method with the parent set identification. We directly encode the candidate parent sets instead of using the paths $\boldsymbol{d}$.

Moreover, we decompose the candidate parent sets $(W_{h,i})_{h=0}^{\lambda_i}$ of each random variable into the form of Cartesian products as follows:

1. Decompose $(W_{h,i})_{h=0}^{\lambda_i}$ into $(W_{h,i} \cap Z_i)_{h=0}^{\lambda_i}, (W_{h,i} \cap (\mathcal{X} \setminus Z_i))_{h=0}^{\lambda_i}$,

2. Remove duplicates in the elements of $(W_{h,i} \cap Z_i)_{h=0}^{\lambda_i}, (W_{h,i} \cap (\mathcal{X} \setminus Z_i))_{h=0}^{\lambda_i}$,

105     3. Store $(W_{h,i} \cap Z_i)_{h=0}^{\lambda_{1,i}}, (W_{h,i} \cap (\mathcal{X} \setminus Z_i))_{h=0}^{\lambda_{2,i}}$ in $(U_{h,i})_{h=0}^{\lambda_{1,i}}, (V_{h,i})_{h=0}^{\lambda_{2,i}}$,

106 where $Z_i \subseteq \cup_{h=0}^{\lambda_i} W_{h,i}, W_{0,i} = U_{0,i} = V_{0,i} = \phi, \lambda_i, \lambda_{1,i}, \lambda_{2,i} \in \mathbb{N} \cup \{0\}$ for all $1 \leq i \leq n$. There
107 is a clear relationship,

$$\{W_{0,i}, \cdots, W_{\lambda_i,i}\} \subseteq \{U \cup V \mid (U, V) \in \{U_{0,i}, \cdots, U_{\lambda_{1,i},i}\} \times \{V_{0,i}, \cdots, V_{\lambda_{2,i},i}\}\}, \quad (8)$$

108 for all $1 \leq i \leq n$. Here, given that the Hamiltonian is a quadratic pseudo-Boolean function, we can
109 represent the score against $U_{h,i} \cup V_{h',i}$ by allocating $U_{h,i}, V_{h',i}$ to two bits on the Hamiltonian. There-
110 fore, it is possible to encode the candidate parent sets into the Hamiltonian using $(U_{h,i})_{h=0}^{\lambda_{1,i}}, (V_{h,i})_{h=0}^{\lambda_{2,i}}$.
111 The number of required bits of the score component of the Hamiltonian is $\sum_{i=1}^{n}(\lambda_{1,i} + \lambda_{2,i})$.

112 **Example 1.** An example of the candidate parent sets in the form of Cartesian products as follows:

$$\mathcal{X} = \{X_1, X_2, X_3, X_4\}, \ \ Z_i = \{X_1, X_2\}, \ \ \lambda_i = 5, \ \ \lambda_{1,i} = 2, \ \ \lambda_{2,i} = 1$$
$$(W_{h,i})_{h=0}^{\lambda_i} = (\phi, \{X_1\}, \{X_1, X_2\}, \{X_3, X_4\}, \{X_1, X_3, X_4\}, \{X_1, X_2, X_3, X_4\}),$$
$$(W_{h,i} \cap Z_i)_{h=0}^{\lambda_i} = (\phi, \{X_1\}, \{X_1, X_2\}, \phi, \{X_1\}, \{X_1, X_2\}),$$
$$(W_{h,i} \cap (\mathcal{X} \setminus Z_i))_{h=0}^{\lambda_i} = (\phi, \phi, \phi, \{X_3, X_4\}, \{X_3, X_4\}, \{X_3, X_4\}),$$
$$(U_{h,i})_{h=0}^{\lambda_{1,i}} = (\phi, \{X_1\}, \{X_1, X_2\}), \ \ (V_{h,i})_{h=0}^{\lambda_{2,i}} = (\phi, \{X_3, X_4\}).$$

113 We optimize $Z_i \subseteq \cup_{h=0}^{\lambda_i} W_{h,i}$ to minimize $\lambda_{1,i} + \lambda_{2,i}$. However, it is often infeasible to search all
114 elements of the power set $\mathcal{P}(\cup_{h=0}^{\lambda_i} W_{h,i})$. Therefore, we heuristically search $Z_i$ adding elements one
by one, as algorithm 1. The computational cost is at most $\mathcal{O}(\lambda_i^3)$ for all $1 \leq i \leq n$.

---

**Algorithm 1** Greedy Candidate Parent Set Decomposition

---

1: **Input:** $(W_{h,i})_{h=0}^{\lambda_i}$ **Output:** $Z$ **Initialize:** $\lambda \leftarrow \lambda_i, Z' \leftarrow \phi, Z \leftarrow \phi$.
2: **for** $d = 1$ to $|\cup_{h=0}^{\lambda_i} W_{h,i}| - 1$ **do**
3:     **for** $X$ in $\cup_{h=0}^{\lambda_i} W_{h,i} \setminus Z$ **do**
4:         **if** $\lambda_{1,i} + \lambda_{2,i} < \lambda$ for $Z_i = Z \cup \{X\}$ **then** $\lambda \leftarrow \lambda_{1,i} + \lambda_{2,i}, Z' \leftarrow Z \cup \{X\}$.
5:     **if** $Z \neq Z'$ **then** $Z \leftarrow Z'$ **else break**

---

115

116 **Example 2.** An example of the bit reduction flow of algorithm 1 is as follows:

$$Z_i = \phi, \lambda_{1,i} = 0, \lambda_{2,i} = 5 : (\phi) \times (\phi, \{X_1\}, \{X_1, X_2\}, \{X_3, X_4\}, \{X_1, X_3, X_4\}, \{X_1, X_2, X_3, X_4\}),$$
$$Z_i = \{X_1\}, \lambda_{1,i} = 1, \lambda_{2,i} = 3 : (\phi, \{X_1\}) \times (\phi, \{X_2\}, \{X_3, X_4\}, \{X_2, X_3, X_4\}),$$
$$Z_i = \{X_1, X_2\}, \lambda_{1,i} = 2, \lambda_{2,i} = 1 : (\phi, \{X_1\}, \{X_1, X_2\}) \times (\phi, \{X_3, X_4\}).$$

## 4   Efficient Conversion of Score-based Bayesian Network Structure Learning

118 We make $(U_{h,i})_{h=0}^{\lambda_{1,i}}, (V_{h,i})_{h=0}^{\lambda_{2,i}}$ correspond to $(p_{h,i})_{h=0}^{\lambda_{1,i}}, (q_{h,i})_{h=0}^{\lambda_{2,i}}$ one-to-one, where $p_{h,i}, q_{h',i} \in \mathbb{B}$
119 for all $0 \leq h \leq \lambda_{1,i}, 0 \leq h' \leq \lambda_{2,i}, 1 \leq i \leq n$. To identify the parent sets, we use the one-to-one
120 correspondence constraint that $\sum_{h=0}^{\lambda_{1,i}} p_{h,i} = \sum_{h=0}^{\lambda_{2,i}} q_{h,i} = 1$ for all $1 \leq i \leq n$. The Hamiltonian
121 consists of the score component, the one-to-one correspondence constraint, and the cycle constraint,

$$H_{\text{total}}^*(\boldsymbol{p}, \boldsymbol{q}, \boldsymbol{r}) \equiv \sum_{i=1}^{n}(H_{\text{score}}^{*(i)}(\boldsymbol{p}_{\cdot,i}, \boldsymbol{q}_{\cdot,i}) + H_{\text{one}}^{*(i)}(\boldsymbol{p}_{\cdot,i}, \boldsymbol{q}_{\cdot,i})) + H_{\text{cycle}}^*(\boldsymbol{p}, \boldsymbol{q}, \boldsymbol{r}), \quad (9)$$

122 where $\boldsymbol{p} = ((p_{h,i})_{h=0}^{\lambda_{1,i}})_{i=1}^{n}, \boldsymbol{q} = ((q_{h,i})_{h=0}^{\lambda_{2,i}})_{i=1}^{n}$. Under the one-to-one correspondence constraint,
123 we can represent the paths among random variables indirectly using $\boldsymbol{p}, \boldsymbol{q}$ without additional auxiliary
124 variables,

$$d_{j,i}^* \equiv \sum_{\substack{1 \leq h \leq \lambda_{1,i} \\ X_j \in U_{h,i}}} p_{h,i} + \sum_{\substack{1 \leq h \leq \lambda_{2,i} \\ X_j \in V_{h,i}}} q_{h,i}, \quad (10)$$

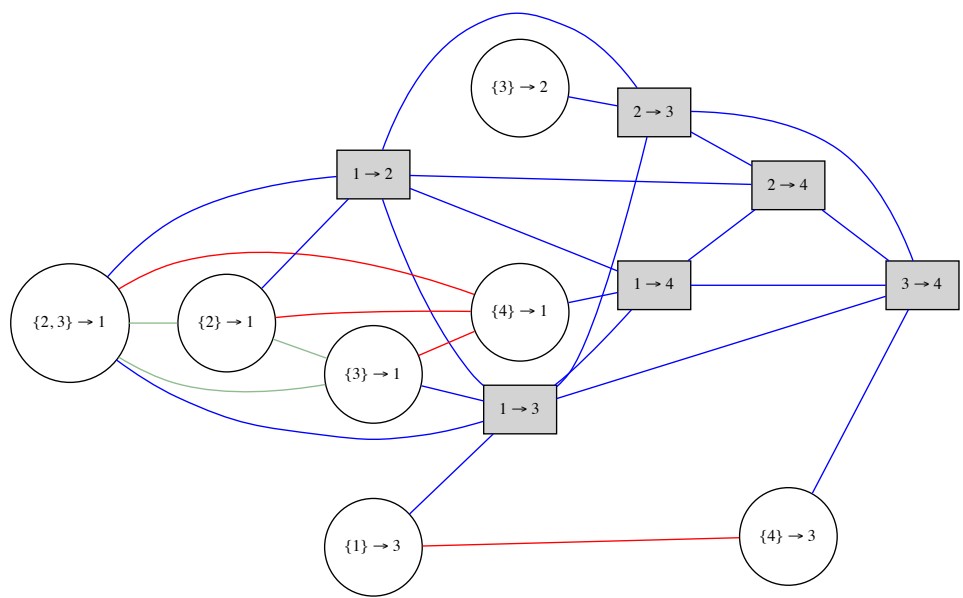

Figure 1: An example of bit allocation for our conversion method. $n = 4, \lambda_{1,1} = 3, \lambda_{2,1} = 1, \lambda_{1,2} = 1, \lambda_{2,2} = 0, \lambda_{1,3} = 1, \lambda_{2,3} = 1, \lambda_{1,4} = 0, \lambda_{2,4} = 0, U_{1,1} = \{2,3\}, U_{2,1} = \{3\}, U_{3,1} = \{2\}, V_{1,1} = \{4\}, U_{1,2} = \{3\}, U_{1,3} = \{1\}, V_{1,3} = \{4\}$. Circle : $\boldsymbol{p}, \boldsymbol{q}$. Square : $\boldsymbol{r}$. Red lines include in the score component of the Hamiltonian, a green line in the one-to-one correspondence constraint, and blue lines in the cycle constraint.

for all $1 \leq j, i \leq n$. Figure 1 is an example of bit allocation using our conversion method. The number of bits required in our conversion method is $\sum_{i=1}^{n}(\lambda_{1,i} + \lambda_{2,i}) + \binom{n}{2}$. Note that we do not directly encode $p_{0,i}, q_{0,i}$ on the Hamiltonian.

**Score Component.** The local score component of the Hamiltonian is

$$H_{\text{score}}^{*(i)}(\boldsymbol{p}_{\cdot,i}, \boldsymbol{q}_{\cdot,i}) \equiv \sum_{h=1}^{\lambda_{1,i}} s_{1,h,i} p_{h,i} + \sum_{h=1}^{\lambda_{2,i}} s_{2,h,i} q_{h,i} + \sum_{h=1}^{\lambda_{1,i}} \sum_{h'=1}^{\lambda_{2,i}} t_{h,h',i} p_{h,i} q_{h',i}, \quad (11)$$

for all $1 \leq i \leq n$. We can get these coefficients by solving simultaneous equations under the one-to-one correspondence constraint, $s_{1,h,i} = -\log S^{(i)}(U_{h,i} \mid \mathcal{D}) + \log S^{(i)}(\phi \mid \mathcal{D}), s_{2,h,i} = -\log S^{(i)}(V_{h,i} \mid \mathcal{D}) + \log S^{(i)}(\phi \mid \mathcal{D}), t_{h,h',i} = -\log S^{(i)}(U_{h,i} \cup V_{h',i} \mid \mathcal{D}) + \log S^{(i)}(U_{h,i} \mid \mathcal{D}) + \log S^{(i)}(V_{h',i} \mid \mathcal{D}) - \log S^{(i)}(\phi \mid \mathcal{D})$.

**One-to-One Correspondence Constraint.** We penalize the connection among bits to select each element from $(U_{h,i})_{h=0}^{\lambda_{1,i}}, (V_{h,i})_{h=0}^{\lambda_{2,i}}$,

$$H_{\text{one}}^{*(i)}(\boldsymbol{p}_{\cdot,i}, \boldsymbol{q}_{\cdot,i}) \equiv \sum_{1 \leq h < h' \leq \lambda_{1,i}} \xi_{1,i} p_{h,i} p_{h',i} + \sum_{1 \leq h < h' \leq \lambda_{2,i}} \xi_{2,i} q_{h,i} q_{h',i}, \quad (12)$$

for all $1 \leq i \leq n$, where the penalty coefficient $0 < \xi_{1,i}, \xi_{2,i} \in \mathbb{R}$. If $\xi_{1,i}, \xi_{2,i}$ is sufficient large, $\sum_{h=0}^{\lambda_{1,i}} p_{h,i} = \sum_{h=0}^{\lambda_{2,i}} q_{h,i} = 1$ is induced indirectly.

**Cycle Constraint.** Compared to eq. (6), the cycle constraint of the Hamiltonian is

$$H_{\text{cycle}}^{*}(\boldsymbol{p}, \boldsymbol{q}, \boldsymbol{r}) \equiv \sum_{1 \leq i < j < k \leq n} \delta_1^* R(r_{i,j}, r_{j,k}, r_{i,k}) + \sum_{1 \leq i < j \leq n} \delta_2^* (d_{i,j}^* r_{i,j} + d_{j,i}^* (1 - r_{i,j})), \quad (13)$$

where the penalty coefficients $0 < \delta_1^*, \delta_2^* \in \mathbb{R}$. By setting $\delta_1^*, \delta_2^*$ appropriately, we can prevent the cycle from occurring.

## 5 Sufficient Lower Bounds of Penalty Coefficients

We demonstrate the sufficient lower bounds of penalty coefficients. The basic idea is that we find the range of penalty coefficients so that the change in return value of the Hamiltonian is negative when the input state changes to the state we desire to induce.

**One-to-One Correspondence Constraint.** We consider to decrease the value of $\sum_{h=1}^{\lambda_{1,i^*}} p_{h,i^*}$ by one until it reaches 1. In the case of $p_{h^*,i^*} = 1, \sum_{h=1}^{\lambda_{1,i^*}} p_{h,i^*} > 1$, it holds that $H_{\text{total}}^*(\boldsymbol{p}, \boldsymbol{q}, \boldsymbol{r}) - H_{\text{total}}^*(\boldsymbol{p}^{(h^*,i^*)}, \boldsymbol{q}, \boldsymbol{r}) \geq \xi_{1,i^*} + s_{1,h^*,i^*} + \sum_{h=1}^{\lambda_{2,i}} t_{h^*,h,i^*} q_{h,i^*}$, where $\boldsymbol{p}^{(h^*,i^*)} = ((p_{h,i}^{(h^*,i^*)})_{h=0}^{\lambda_{1,i}})_{i=1}^n$, and $p_{h,i}^{(h^*,i^*)} = 0$ if $(h,i) = (h^*,i^*)$, $p_{h,i}^{(h^*,i^*)} = p_{h,i}$ otherwise. Considering the case where $\boldsymbol{p}$ and $\boldsymbol{q}$ are swapped in the above, if $\xi_{1,i}, \xi_{2,i}$ satisfy that

$$\max_{0 \leq h \leq \lambda_{1,i}} \left( -s_{1,h,i} - \sum_{h'=1}^{\lambda_{2,i}} \min\{0, t_{h,h',i}\} \right) < \xi_{1,i}, \tag{14}$$

$$\max_{0 \leq h \leq \lambda_{2,i}} \left( -s_{2,h,i} - \sum_{h'=1}^{\lambda_{1,i}} \min\{0, t_{h',h,i}\} \right) < \xi_{2,i}, \tag{15}$$

for all $1 \leq i \leq n$, then the grand state does not violate the one-to-one correspondence constraint. The computational cost to obtain the left side of eq. (14) and eq. (15) is at most $\mathcal{O}(\lambda_{1,i}\lambda_{2,i})$ for all $1 \leq i \leq n$.

**Cycle Constraint.** We consider four patterns of $(r_{i^*,j^*}, d_{j^*,i^*}^*, d_{i^*,j^*}^*)$ violating the consistency constraint. It is assumed that $X_{j^*} \in U_{h^*,i^*}, X_{j^*} \notin U_{h^{**},i^*} \subset U_{h^*,i^*}, p_{h^*,i^*} = 1, p_{h^{**},i^*} = 0$. In the case of $(0,1,0)$, it holds that $H_{\text{total}}^*(\boldsymbol{p}, \boldsymbol{q}, \boldsymbol{r}) - H_{\text{total}}^*(\boldsymbol{p}, \boldsymbol{q}, \boldsymbol{r}^{(i^*,j^*)}) \geq \delta_2^* - (n-2)\delta_1^*$, where $\boldsymbol{r}^{(i^*,j^*)} = (r_{i,j}^{(i^*,j^*)})_{1 \leq i < j \leq n}$, and $r_{i,j}^{(i^*,j^*)} = 1 - r_{i,j}$ if $(i,j) = (i^*,j^*)$, $r_{i,j}^{(i^*,j^*)} = r_{i,j}$ otherwise. Similarly, it is possible to consider the case of $(1,0,1)$. In the case of $(0,1,1)$, it holds that $H_{\text{total}}^*(\boldsymbol{p}, \boldsymbol{q}, \boldsymbol{r}) - H_{\text{total}}^*(\boldsymbol{p}^{(h^*,h^{**},i^*)}, \boldsymbol{q}, \boldsymbol{r}) \geq \delta_2^* + s_{1,h^*,i^*} + \sum_{h=1}^{\lambda_{2,i}} t_{h^*,h,i^*} q_{h,i^*} - s_{1,h^{**},i^*} - \sum_{h=1}^{\lambda_{2,i}} t_{h^{**},h,i^*} q_{h,i^*}$, where $\boldsymbol{p}^{(h^*,h^{**},i^*)} = ((p_{h,i}^{(h^*,h^{**},i^*)})_{h=0}^{\lambda_{1,i}})_{i=1}^n$, and $p_{h,i}^{(h^*,h^{**},i^*)} = 0$ if $(h,i) = (h^*,i^*)$, $p_{h,i}^{(h^*,h^{**},i^*)} = 1$ if $(h,i) = (h^{**},i^*)$, $p_{h,i}^{(h^*,h^{**},i^*)} = p_{h,i}$ otherwise. Similarly, it is possible to consider the case of $(1,1,1)$. These results suggest the relationship of $\delta_1^*, \delta_2^*$ to induce the consistency constraint. Here, based on theorem 1, we consider a strategy to repeat picking up one element from $\boldsymbol{r}$ and switching its value until $H_{\text{trans}}(\boldsymbol{r}) = 0$. It is assumed that $H_{\text{trans}}(\boldsymbol{r}) > H_{\text{trans}}(\boldsymbol{r}^{(i^*,j^*)})$. In the case of $(1,1,0)$, it holds that $H_{\text{total}}^*(\boldsymbol{p}, \boldsymbol{q}, \boldsymbol{r}) - H_{\text{total}}^*(\boldsymbol{p}^{(h^*,h^{**},i^*)}, \boldsymbol{q}, \boldsymbol{r}^{(i^*,j^*)}) \geq \delta_1^* + s_{1,h^*,i^*} + \sum_{h=1}^{\lambda_{2,i}} t_{h^*,h,i^*} q_{h,i^*} - s_{1,h^{**},i^*} - \sum_{h=1}^{\lambda_{2,i}} t_{h^{**},h,i^*} q_{h,i^*}$. Similarly, it is possible to consider the case of $(0,0,1)$. In the case of $(1,0,0)$ or $(0,0,0)$, it holds that $H_{\text{total}}^*(\boldsymbol{p}, \boldsymbol{q}, \boldsymbol{r}) - H_{\text{total}}^*(\boldsymbol{p}, \boldsymbol{q}, \boldsymbol{r}^{(i^*,j^*)}) \geq \delta_1^*$. These results suggest the lower bound of $\delta_1^*$ to induce the topological order constraint. Considering the case where $\boldsymbol{p}$ and $\boldsymbol{q}$ are swapped in the above, if $\delta_1^*, \delta_2^*$ satisfy that

$$\max_{1 \leq i \leq n} \max\{\eta_{1,i}, \eta_{2,i}\} < \delta_1^* < \frac{\delta_2^*}{n-2}, \tag{16}$$

$$\eta_{1,i} \equiv \max_{\substack{1 \leq j \leq n}} \max_{\substack{0 \leq h \leq \lambda_{1,i} \\ X_j \in U_{h,i}}} \max_{\substack{0 \leq h' \leq \lambda_{1,i} \\ X_j \notin U_{h',i} \subset U_{h,i}}} \max_{0 \leq h'' \leq \lambda_{2,i}} (-s_{1,h,i} - t_{h,h'',i} + s_{1,h',i} + t_{h',h'',i}),$$

$$\eta_{2,i} \equiv \max_{\substack{1 \leq j \leq n}} \max_{\substack{0 \leq h \leq \lambda_{2,i} \\ X_j \in V_{h,i}}} \max_{\substack{0 \leq h' \leq \lambda_{2,i} \\ X_j \notin V_{h',i} \subset V_{h,i}}} \max_{0 \leq h'' \leq \lambda_{1,i}} (-s_{2,h,i} - t_{h'',h,i} + s_{2,h',i} + t_{h'',h',i}),$$

for $n \geq 3$, then the grand state does not violate the cycle constraint under the one-to-one correspondence constraint. The computational cost to obtain the left side of eq. (16) is at most $\mathcal{O}(\sum_{i=1}^n n\lambda_{1,i}\lambda_{2,i}(\lambda_{1,i} + \lambda_{2,i}))$.

**Theorem 1.** *If it holds that $H_{\text{trans}}(\boldsymbol{r}) \equiv \sum_{1 \leq i < j < k \leq n} R(r_{i,j}, r_{j,k}, r_{i,k}) > 0$, then there exists at least one index pair $1 \leq i^* < j^* \leq n$ which satisfy $H_{\text{trans}}(\boldsymbol{r}) > H_{\text{trans}}(\boldsymbol{r}^{(i^*,j^*)})$, where $R(r_1, r_2, r_3) = r_1 r_2 (1-r_3) + (1-r_1)(1-r_2)r_3$ for all $r_1, r_2, r_3 \in \mathbb{B}$, $\boldsymbol{r} = (r_{i,j})_{1 \leq i < j \leq n} \in \mathbb{B}^{\binom{n}{2}}$, $\boldsymbol{r}^{(i^*,j^*)} = (r_{i,j}^{(i^*,j^*)})_{1 \leq i < j \leq n}$, and $r_{i,j}^{(i^*,j^*)} = 1 - r_{i,j}$ if $(i,j) = (i^*,j^*)$, $r_{i,j}^{(i^*,j^*)} = r_{i,j}$ otherwise.*

Table 1: The benchmark networks from Bayesian network repository.

| Name | $n$ | $m$ | $\sum_{i=1}^{n} |\Pi_i|$ | $\sum_{i=1}^{n} \beta_i(\gamma_i - 1)$ | $\sum_{i=1}^{n} \lambda_i$ * | | |
|---|---|---|---|---|---|---|---|
| | | | | | $N = 100$ | $N = 1000$ | $N = 10000$ |
| insurance | 27 | 3 | 52 | 984 | 353 | 883 | 4036 |
| water | 32 | 5 | 66 | 10083 | 165 | 216 | 735 |
| alarm | 37 | 4 | 46 | 509 | 1829 | 2272 | 9081 |
| barley | 48 | 4 | 84 | 114005 | 181 | 310 | 1552 |
| hailfinder | 56 | 4 | 66 | 2656 | 144 | 692 | 4277 |
| hepar2 | 70 | 6 | 123 | 1453 | 4837 | 665 | 4782 |

* The average for 10 simulated datasets.

*Proof.* It does not lose the generality by considering the case of $(r_{1,2}, r_{2,3}, r_{1,3}) = (1, 1, 0)$. Here, it holds that $R(r_{1,2}, r_{2,3}, r_{1,3}) - R(1 - r_{1,2}, r_{2,3}, r_{1,3}) + R(r_{1,2}, r_{2,3}, r_{1,3}) - R(r_{1,2}, 1 - r_{2,3}, r_{1,3}) + R(r_{1,2}, r_{2,3}, r_{1,3}) - R(r_{1,2}, r_{2,3}, 1 - r_{1,3}) = 3$. Additionally, it holds that $R(r_{1,2}, r_{2,i}, r_{1,i}) - R(1 - r_{1,2}, r_{2,i}, r_{1,i}) + R(r_{2,3}, r_{3,i}, r_{2,i}) - R(1 - r_{2,3}, r_{3,i}, r_{2,i}) + R(r_{1,3}, r_{3,i}, r_{1,i}) - R(1 - r_{1,3}, r_{3,i}, r_{1,i}) = 0$ for all $3 < i$. Therefore, it holds that $H_{\text{trans}}(\boldsymbol{r}) - H_{\text{trans}}(\boldsymbol{r}^{(1,2)}) + H_{\text{trans}}(\boldsymbol{r}) - H_{\text{trans}}(\boldsymbol{r}^{(2,3)}) + H_{\text{trans}}(\boldsymbol{r}) - H_{\text{trans}}(\boldsymbol{r}^{(1,3)}) = 3$. From this result, it holds that $H_{\text{trans}}(\boldsymbol{r}) - H_{\text{trans}}(\boldsymbol{r}^{(i^*, j^*)}) > 0$ for at least one index pair $(i^*, j^*) \in \{(1, 2), (2, 3), (1, 3)\}$. □

## 6 Experimental Results

To validate the performance of our approach, we use 10 simulated datasets for each instance size $N = 100, 1000, 10000$ and each benchmark network. The benchmark networks are discrete networks from Bayesian network repository [1]. The score function is the BDeu score with $\alpha = 1$. It is often infeasible to identify exact candidate parent sets by searching the power set $\mathcal{P}(\mathcal{X} \setminus \{X_i\})$ in a realistic timeframe. We use the candidate parent sets from algorithm 2. Note that the candidate parent sets depend on the heuristic search algorithms, but we do not focus on their performance in this study. Table 1 displays the information of benchmark networks. The code to replicate each experiment in this paper is available [2].

---

**Algorithm 2** Greedy Candidate Parent Set Identification

---

1: **Input:** $\mathcal{D}, i, m$ **Output:** $\mathcal{L}$ **Initialize:** $\mathcal{L} \leftarrow \{\phi\}, \mathcal{L}' \leftarrow \{\phi\}, \mathcal{L}'' \leftarrow \phi$
2: **for** $d = 1$ to $m$ **do**
3:     **for** $W$ in $\mathcal{L}'$ **do**
4:         **for** $X$ in $\mathcal{X} \setminus \{X_i\} \setminus W$ **do**
5:             **if** $S_i(W' \,|\, \mathcal{D}) < S_i(W \cup \{X\} \,|\, \mathcal{D})$ for all $W' \subset W \cup \{X\}, W' \in \mathcal{L}$ **then**
6:                 $\mathcal{L}'' \leftarrow \mathcal{L}'' \cup \{W \cup \{X\}\}$.
7:     **if** $\mathcal{L}'' \neq \phi$ **then** $\mathcal{L} \leftarrow \mathcal{L} \cup \mathcal{L}'', \mathcal{L}' \leftarrow \mathcal{L}'', \mathcal{L}'' \leftarrow \phi$ **else break**
8: **for** $W$ in $\mathcal{L}$ **do**
9:     **if** there exist $W' \subset W$ that satisfies $S_i(W \,|\, \mathcal{D}) \leq S_i(W' \,|\, \mathcal{D})$ **then** $\mathcal{L} \leftarrow \mathcal{L} \setminus \{W\}$.

---

### 6.1 Number of Required Bits for Score Component

In comparison to the existing method [O'Gorman et al., 2014], we reduce the number of required bits for the score component by encoding the candidate parent sets directly. While $\sum_{i=1}^{n} \lambda_i$ candidate parent sets is encoded in our approach, $n(n-1)$ paths plus at most $\mathcal{O}(n(n-1)^{\frac{m}{2}})$ auxiliary variables for $m > 2$ in the existing method. The left side of table 2 shows the reduction rate of the number of required bits for the score component. Moreover, we reduce the number of required bits for the score component to $\sum_{i=1}^{n} (\lambda_{1,i} + \lambda_{2,i})$ by decomposing the candidate parent sets in the form of Cartesian

[1] https://www.bnlearn.com/bnrepository/
[2] See supplemental material.

Table 2: The reduction rate of the number of required bits for score component.

| Name | $\sum_{i=1}^{n} \lambda_i \, / \, n(n-1)^{\frac{m}{2}}$ * | | | $\sum_{i=1}^{n}(\lambda_{1,i} + \lambda_{2,i}) \, / \sum_{i=1}^{n} \lambda_i$ * | | |
|---|---|---|---|---|---|---|
| | $N = 100$ | $N = 1000$ | $N = 10000$ | $N = 100$ | $N = 1000$ | $N = 10000$ |
| insurance | 0.09873 | 0.24677 | 1.12742 | 0.61367 | 0.47285 | 0.32476 |
| water | 0.00097 | 0.00126 | 0.00429 | 0.72680 | 0.70588 | 0.44014 |
| alarm | 0.03814 | 0.04738 | 0.18938 | 0.45332 | 0.35537 | 0.21617 |
| barley | 0.00171 | 0.00292 | 0.01464 | 0.76717 | 0.75538 | 0.54149 |
| hailfinder | 0.00085 | 0.00409 | 0.02525 | 0.82773 | 0.60178 | 0.33365 |
| hepar2 | 0.00021 | 0.00003 | 0.00021 | 0.49694 | 0.63346 | 0.31284 |

\* The average ratio for 10 simulated datasets.

Table 3: The number of required bits for fully coupled and nearest neighbor annealing processors.

| Name | $\sum_{i=1}^{n}(\lambda_{1,i} + \lambda_{2,i}) + \binom{n}{2}$ * | | | $\sum_{i=1}^{n}(\lambda_{1,i} + \lambda_{2,i})(\lambda_{1,i} + \lambda_{2,i} + 1) + \binom{n}{2}$ * | | |
|---|---|---|---|---|---|---|
| | $N = 100$ | $N = 1000$ | $N = 10000$ | $N = 100$ | $N = 1000$ | $N = 10000$ |
| insurance | 566 | 767 | 1661 | 3375 | 9023 | 85482 |
| water | 613 | 648 | 820 | 1434 | 1720 | 5881 |
| alarm | 1489 | 1472 | 2628 | 36761 | 27985 | 169004 |
| barley | 1247 | 1362 | 1968 | 6758 | 3446 | 24796 |
| hailfinder | 1659 | 1957 | 2967 | 2212 | 7084 | 80578 |
| hepar2 | 4777 | 2836 | 3910 | 449916 | 9164 | 136939 |

\* The average ratio for 10 simulated datasets.

products. The right side of table 2 shows that algorithm 1 reduces the number of required bits for the score component although there is some variation among the networks.

## 6.2 Selection of Annealing Processor

From the following discussion, the Fujitsu digital annealer is suitable for our approach from the viewpoint of bit capacity.

**Fully Connected Type.** To the best of our knowledge, the bit capacity of the Fujitsu digital annealer is the largest in fully coupled annealing processors. The second generation Fujitsu digital annealer can deal with problems on a scale of 8192 bits [Matsubara et al., 2020]. The left side of table 3 shows that it is possible to encode all the logical conversion results for benchmark networks to the circuit of the digital annealer within bit capacity.

**Nearest Neighbor Type.** The number of additional bits required for minor embedding depends on the design of the hardware graphs. Oku et al. 2019 proposed a heuristic minor embedding algorithm for the Hitachi CMOS annealing machine [Masanao et al., 2010]. Using this algorithm, the number of required physical spins when embedding a fully connected graph is $I^2 + I$ for $I$ variables. The conversion method proposed in this study has $n$ local fully connected graphs on $\boldsymbol{p}, \boldsymbol{q}$. Therefore, the number of required physical spins must be at least $\sum_{i=1}^{n}(\lambda_{1,i} + \lambda_{2,i})(\lambda_{1,i} + \lambda_{2,i} + 1) + \binom{n}{2}$. From the right side of table 3, it is currently infeasible to encode logical conversion results for at least some networks to the circuit of CMOS annealing machine within its 102400 nodes [Sugie et al., 2021]. As far as we know, the bit capacity of the Hitachi CMOS annealing machine is the largest in nearest neighbor annealing processors.

## 6.3 Score Maximization

We demonstrate the performance of Fujitsu digital annealer for score-based Bayesian network structure learning using the conversion results of $N = 10000$ simulated datasets. The running time for each simulated dataset is 6000 [s].

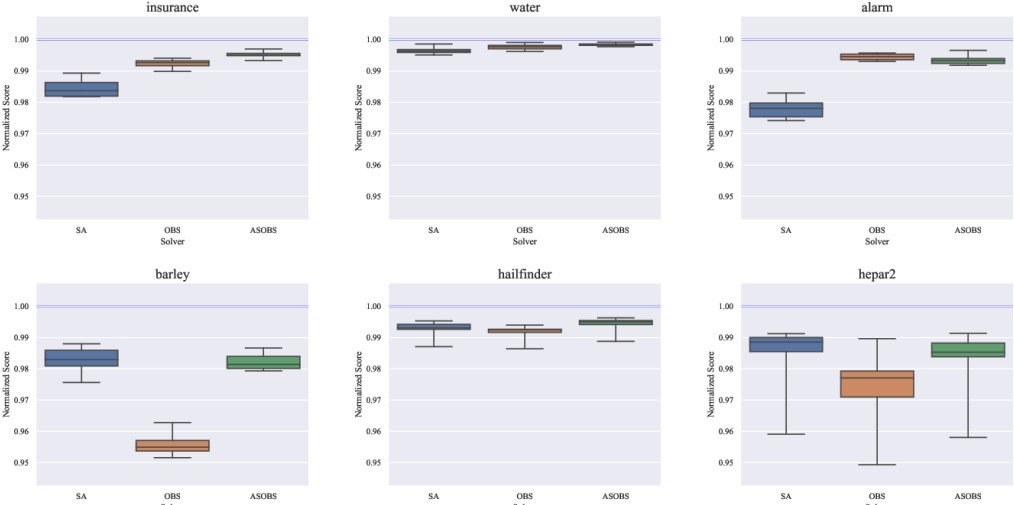

Figure 2: Results of score maximization by the baseline algorithms. For each simulated dataset and each baseline algorithm, we normalized $\sum_{i=1}^{n}(\log S^{(i)}(\Pi_i|\mathcal{D}) - \log S^{(i)}(\phi|\mathcal{D}))$ by dividing it by the corresponding value of the Fujitsu digital annealer. In this experiment, we used the second-generation Fujitsu digital annealer. SA : simulated annealing, OBS : ordering-based search, ASOBS : acyclic selection ordering-based search.

**Baselines.** We compare the results obtained by the digital annealer with those of three heuristic algorithms. One algorithm is the simulated annealing algorithm [Heckerman et al., 1995b] with a QUBO same as the one encoded into the digital annealer. Other algorithms are the ordering space search algorithms, i.e., ordering-based search and acyclic selection ordering-based search. For a fair comparison, the running time of the simulated annealing algorithm for each simulated dataset is 6000 [s] and that of the ordering space search algorithms is 6000 [s] plus the running time of algorithm 1. The computing environment is Microsoft Windows 10 Pro, 3.6 GHz Intel Core i9 processor, and 64 GB memory.

**Result.** Figure 2 shows that the digital annealer is better than all the baselines for all the simulated datasets from all the benchmark networks.

# 7 Conclusion

We proposed a novel approach of converting a score-based Bayesian network structure learning into QUBO. The essence of this approach lies in reducing the number of required bits through the advanced identification of candidate parent sets and their representation as Cartesian products. The Fujitsu digital annealer with our conversion method improved the BDeu score for 27 to 70 variables benchmark networks over existing methods. The bit capacity limitation of annealing processor is being relaxed rapidly [3]. Though our approach is still a disadvantage for larger-scale networks, we expect that our proposed algorithms will be effectively applied to larger-scale score-based Bayesian network structure learning in the near future.

**Potential Negative Societal Impacts.** The development of annealing processor technology could have an impact on various industry fields. However, the number of companies that have commercialized the API usage of annealing processors is still small. Therefore, there is a concern that the market of annealing processors will not work well and the disparities among stakeholders will be widen. Researchers are required to properly evaluate the value of technology and communicate it to the business side.

---

[3]Fujitsu announced that they achieved a megabit-class performance with digital annealer

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
