# OpenReview forum: "Bayesian Network Structure Learning using Digital Annealer"
_NeurIPS.cc/2021/Conference — NeurIPS 2021 Submitted_

### Official Review · Reviewer_Yh3Z · 2021-07-13

**Rating:** 3
**Confidence:** 5

**Summary:**

The paper concerns a method for learning BNs from discrete data. A
score-based approach is taken where the score is BDeu with effective
sample size = 1. Table 1 shows the learning problems attempted
here. The steps are: (1) Choose a limit on parent set size (here
simulated data is used, and the chosen size is that of the biggest
parent set in the true BN). (2) Generate candidate parent sets
(CPSs). Since the authors claim that it is "infeasible to identify
exact candidate parent sets by searching the power set P(X \ {X i })
in a realistic timeframe." a greedy approach it taken (Algorithm
2). (3) Decompose the CPSs into Cartesian products (another greedy
algorithm is used - Algorithm 1). (4) Use a digital annealer to search
for a high-scoring BN. (Comparisions with some other algorithms are
also done).


**Limitations And Societal Impact:**

They addressed potential negative societal impacts but I did not understand the point they were making.

**Main Review:**

As the authors correctly point out BN learning is NP-hard. They also
state that "the standard methodology is using heuristic
approaches". But for 4 of the 6 problems considered here it is
possible to find a provably optimal (maximal score) BN in a reasonable
amount of time.

The authors are to be commended for their supplementary material which
makes it easy to examine their experiments. I used their code (the R
script) to simulate a dataset of size 10000 from the insurance, water,
alarm and hailfinder. Note that 10000 is the largest size of dataset
considered by the authors. I then used gobnilp (C version from public
git repo commit e60ef14, linked to SCIP version 7.0.1 with CPLEX
20.1.0.0 as the LP solver, on a 2.7 GHz laptop) to find optimal BNs
using the same parent set size limits as the authors. The scores of
these optimal BNs and time taken are as follows:

insurance palim=3
BN score is -132686.365245
real	0m26.157s

water palim=5
BN score is -128765.561813
real	0m16.798s

alarm palim=4
BN score is -105196.063958
real	1m50.351s

hailfinder palim=4
BN score is -497399.495910
real	6m18.028s

These times include the time that gobnilp takes to internally
construct candidate parent sets. (gobnilp does not use a greedy
approach to do this to ensure that the generated candidate parent sets
contain those needed for an optimal BN, subject to the given parent
set size limit.) However, both barley(palim=4) and hepar2(palim=6) are
too hard for gobnilp: it fails to even get past the stage of
generating candidate parent sets.

Perhaps some other exact method could solve them, I don't know. But
let's just assume that these two cases are too hard for any 'exact'
(provably optimal) learning. It could be that the authors' method is a
good choice for a heuristic approach. The first step of the authors'
method is to generate a set of candidate parent sets using Algorithm
2. So I used that algorithm to create a set of candidate parent sets
for the barley and hepar2 cases. This takes a while (for barley it
took "real 198m33.421s" to do using the authors' supplied Julia script).

After converting these scores to a format gobnilp understands they
were sent to gobnilp, which, in both cases, quickly found an optimal
BN for the given candidate parent sets:

from barley CPS
BN score is -508892.921268
real	1m26.909s

from hepar2 CPS
BN score is -326309.949418
real	0m11.401s

So in all 6 of the experiments the authors conducted a superior method
is available. Part of the authors' method (greedy generation of
candidate parent sets) could be a useful contribution but I don't
think the current paper provides evidence for that. As it happens if
one lowers the parent set size limit for hepar2 from 5 to 3 and uses
gobnilp, we get a slightly higher scoring BN than that found using the
candidate parent sets greedily generated by the authors' Algorithm 2.

barley, palim=3
BN score is -326303.018128
real	0m42.231s

The current paper contains some interesting ideas and it is reasonable
to explore the value of using an annealing processor to conduct a
heuristic search for a high-scoring BN. However, the authors need to
focus on problems where exact search is infeasible. They also need
to show (empirically) that, given the candidate parent sets they
generate with their Algorithm 2, the annealing approach is a superior
approach to other existing methods which can take a set of candidate
parent sets as input. Some theoretical analysis of 'what is lost' by
the greedy approach of Algorithm 2 would also be welcome.

**Time Spent Reviewing:**

4

---

> ### Author Response · Authors · 2021-08-09
> **Author Response**
>
> We are grateful for showing us the experimental results by exact algorithms. These results are very helpful for evaluating the score by the digital annealer. The exact score seems to be almost same at the digital annealer. Would you confirm "barley, palim=3 BN score is -326303.018128 real 0m42.231s" ? Is it the result of hepar2 ?
>
> We apologize that the scope of contribution is unclear in our manuscript. Our main claim is that our proposed method is superior to the state of arts for the heuristic structure learning after identifying candidate parent sets.
>
> First, Algorithm 2 is not the state of art as you point out. In Algorithm 2, a candidate parent set in the next size does not emerge if all subsets of it do not equal to any candidate parent set in the current size. For instance, it is infeasible to obtain $\{X_1, X_2, X_3\}$ if none of $\{X_1, X_2\}, \{X_2, X_3\}, \{X_3, X_1\}$ correspond to a candidate parent set. However, it is expected for such a case to decrease as $N$ increases. The average ratio on 10 simulated datasets ($N = 10,000$) of the number of candidate parent sets from Algorithm 2 and exact algorithm is insurance : 0.9399 (N = 100), 0.9985 (N = 1000), 0.9978 (N = 10000), water : 0.7827 (N = 100), 0.9773 (N = 1000), 0.9999 (N = 10000), alarm : 0.7537 (N = 100), 0.9891 (N = 1000), 1.0000 (N = 10000). Though Algorithm 2 is not the state of art, it is very simple. That is why we use Algorithm 2 in this experiment to evaluate the performance of the structure learning after identifying candidate parent sets.
>
> Second, the 6 networks in our experiment can be optimized exactly, which is also you point out. Therefore, the experimental results with the 6 networks do not show the superiority of our heuristic method directly. However, from the comparison to the space of orderings [Teyssier and Koller, 2005] [Scanagatta et al, 2015], our experiment provide the indirect evidence to improve the performance of structure learning with larger networks. Due to our budget constraints, we targeted only 6 small networks that can be encoded within the 8192 bits of the second generation Fujitsu digital annealer. However, the third generation of Fujitsu digital annealer has the 100,000 bit capacity. Moreover, Fujitsu announced that they achieved a megabit-class performance with digital annealer. If a megabit-class digital annealer is available to us, we will be able to encode networks with a thousand variables into the circuit of digital annealer.
>
> We look forward to hearing from you regarding our submission. We would be glad to respond to any further questions and comments that you may have.

---

> > ### Comment · Reviewer_Yh3Z · 2021-08-30
> > **response to author response**
> >
> > It is good to hear that the exact score is apparently close to the score achieved by your algorithm (although there is an issue of what counts as "close"). To answer your 1st question: yes, there was a typo in my review, the result you mention was for hepar2 not barley - apologies.
> >
> > I do not think you have established your "main claim" as described in the 2nd para of your response. I think my review showed that it at least one existing algorithm did better when taking candidate parent sets (CPSs) as input. (I suspect there are a number of superior algorithms, but I think one is enough for our current purposes.)
> >
> > Algorithm 2 is a heuristic approach for identifying CPSs. Using a heuristic approach is quite reasonable since an "exact" approach could be infeasible for big problems. Scanagatta et al use one - it would be good to compare to it, our just use the method of Scanagatta et al.
> >
> > It may well be the case that your approach will prove to be a good option for big problems, but we don't have evidence for that at present.

---

### Official Review · Reviewer_GCo9 · 2021-07-15

**Rating:** 5
**Confidence:** 4

**Summary:**

The paper considers the problem of structure learning in Bayesian networks but on alternative hardware to the classic von Neumann computers. In particular, a digital annealer is considered and consequently the structure learning problem is encoded into a quadratic unconstrained binary optimisation problem (QUBO). Digital annealers are known to be very good at (approximately) solving these kinds of optimisation problems. Experimental results on standard Bayesian networks demonstrate the effectiveness of the proposed approach compared with standard methods that run on classical hardware.

**Ethical Concerns:**

There are no ethical concerns.

**Ethics Review Area:**

["I don’t know"]

**Limitations And Societal Impact:**

The limitations are discussed in the paper and there is no potential negative societal impact of this work.

**Main Review:**

The paper is fairly well written and organised. The quality of the presentation is overall fairly good. However, there are a few issues that should be addressed in order to improve the presentation.

1. The notation used throughout the paper, especially when describing the encoding of the Hamiltonian is quite dense and it makes it very difficult to follow and therefore appreciate the contribution. More specifically, I think it should be simplified in some way but more importantly the presentation needs to be accompanied by clearly described examples of the encodings. Figure 1 is kind of confusing as it stands now. I think it would also be helpful to include an example of a digital annealer architecture and how the Hamiltonian/encoding maps to it.

2. In the experimental section, the baselines considered for comparison (ie, structure learning algorithms that run on classical hardware) are not amongst the best performing ones. For instance, there are more recent algorithms that scale to much larger instances and are able to produce superior quality solutions. See for example those introduced by [Scanagatta el al, 2015] or [de Campos et al, 2011].

[Scanagatta et al, 2015] M. Scanagatta, C. De Campos, G. Corani. M. Zaffalon. Learning Bayesian networks with thousands of variables. NeurIPS 2015.

[de Campos et al, 2011] C. De Campos, Q. Ji. Efficient structure learning of Bayesian networks using constraints. JMLR 12:663-689, 2011.


**Time Spent Reviewing:**

3 hours

---

> ### Author Response · Authors · 2021-08-09
> **Author Response**
>
> We appreciate your insightful comment.
>
> The 6 networks in our experiment can be optimized exactly [de Campos et al, 2011]. Therefore, the experimental results with the 6 networks do not show the superiority of our heuristic method directly. However, from the comparison to the acyclic selection ordering-based search [Scanagatta et al, 2015], our experiment provide the indirect evidence to improve the performance of structure learning with larger networks.
> Due to our budget constraints, we targeted only 6 small networks that can be encoded within the 8192 bits of the second generation Fujitsu digital annealer. However, the third generation of Fujitsu digital annealer has the 100,000 bit capacity. Moreover, Fujitsu announced that they achieved a megabit-class performance with digital annealer. If a megabit-class digital annealer is available to us, we will be able to encode networks with a thousand variables into the circuit of digital annealer.
>
> We will make the clear connection between the section 2.3 and the section 4. Moreover, we will rearrange Figure 1 to represent the relationship between the digital annealer architecture and our conversion method.
>
> We look forward to hearing from you regarding our submission. We would be glad to respond to any further questions and comments that you may have.

---

### Official Review · Reviewer_9dre · 2021-07-16

**Rating:** 4
**Confidence:** 2

**Summary:**

**N.B.** I have reviewed a previous version of this paper. This version does not address many of the main concerns I raised in that review, so I will mostly repeat those.

In this work, the authors formulate Bayesian network structure learning (BNSL) as a quadratic unconstrained binary optimization (QUBO) problem. The main contribution of the work is a novel encoding of BNSL as a QUBO respecting the bit limitation of a particular digital annealer platform. Experiments on standard benchmarks show the formulation leads to mildly higher likelihoods compared to baseline methods.

**Ethical Concerns:**

No ethical concerns.

**Limitations And Societal Impact:**

--- Quality

While I appreciate that the main contribution of this work is the BNSL formulation for the Fujitsu annealer, the paper also claims to make meaningful improvements in solving BNSL compared to state of the art. Local search algorithms are not state of the art with the long time frame and relatively small networks here; the score can be exactly optimized by a large number of techniques.

The experiments themselves do not really explore the cases where the formulation and encoding is effective and where it is not. For example, the size of the encoding is dependent on the number of candidate parent sets and how they can be decomposed. It is well known [de Campos and Ji, JMLR 2011; among others] that the number of candidate parent sets is dependent on the number of instances in a dataset as well as the choice of scoring function, among other factors. One possible deeper experiment could explore how these factors affect the QUBO formulation.

--- Clarity

The paper is very difficult to follow for a non-expert. Additional explanations, intuition, context, etc., about what the various algorithms and equations mean would be very helpful for reaching a wider audience.

--- Significance

The significance of the work is limited since the authors do not attempt to connect their formulation of BNSL (such as the Cartesian product approach for representing candidate parent sets) to other work in the area. Similarly, it is not clear if the proposed strategies for quadratization for BNSL would be relevant for other problems, even highly related ones such as learning Markov networks.

--- Societal impact

Annealing processors may indeed have some sort of impact on computing.

**Main Review:**

--- Originality

The main contribution of this work is the specific formulation and encoding of BNSL as a QUBO. In particular, a decomposition of parent sets to a suitable form is given, as well as a way to avoid cycles in the BN structure.

--- Quality

The technical quality of the work is somewhat limited. For example, it is unclear if the size of the encoding provided here is a tight lower bound on the size of an encoding of this formulation of BNSL.

While it is unclear what versions of the order-based searchers were used (with/without restarts, tabu list, etc.), and local search algorithms for the time frame and networks of this size are not state of the art, the BNSL baselines are an improvement compared to the previous version.

--- Clarity

As a non-expert in such annealing approaches, and especially the Hamiltonian formulations, I found this paper extremely difficult to follow. It is very notation heavy, and very little intuition, description, or context is given for the equations and algorithms.

--- Significance

The work shows that the proposed formulation of BNSL as a QUBO using the fully connected DA architecture has advantages compared to other formulations of BNSL for the nearest-neighbor DA architecture. However, the (updated) version of the paper does not provide any hints as to how the work may be of interest to either the broader BNSL community or for other DA/QUBO problems.

**Time Spent Reviewing:**

1.5

---

> ### Author Response · Authors · 2021-08-09
> **Author Response**
>
> We are grateful for the second time comment.
>
> We currently consider to apply our proposed method to other QUBO problems. One of them is the multiple-choice knapsack problem [Kellerer et al, 2004]. It is easy to replace the candidate parent sets of each variable to the candidate item sets of each knapsack and to encode the constraint to avoid the duplication of items among knapsacks instead of the cycle constraint. Note that the Cartesian product approach will contribute to reduce the required bits in this case as well as the Bayesian structure learning. On the contrary, the Cartesian product approach is directly not useful for learning chordal Markov network [Kangas el al, 2014]. However, it is possible to encode the chordal Markov network into the digital annealer.  The details of Markov networks into QUBO are still being worked out, but the basic concept is simple. We encode candidate variable sets in each clique into the circuit of the digital annealer. The couplers represent the likelihood corresponding to the intersection of two adjacent cliques. The required bits is $\mathcal{O}(\frac{n^{l+1}}{l})$ bits for n random variables and a maximum clique size l.
>
> Local search algorithm is not the state of art. In Algorithm 2, a candidate parent set in the next size does not emerge if all subsets of it do not equal to any candidate parent set in the current size. For instance, it is infeasible to obtain $\{X_1, X_2, X_3\}$ if none of $\{X_1, X_2\}, \{X_2, X_3\}, \{X_3, X_1\}$ correspond to a candidate parent set. However, it is expected for such a case to decrease as $N$ increases. The average ratio on 10 simulated datasets ($N = 10,000$) of the number of candidate parent sets from Algorithm 2 and exact algorithm is insurance : 0.9399 (N = 100), 0.9985 (N = 1000), 0.9978 (N = 10000), water : 0.7827 (N = 100), 0.9773 (N = 1000), 0.9999 (N = 10000), alarm : 0.7537 (N = 100), 0.9891 (N = 1000), 1.0000 (N = 10000). Despite Algorithm 2 is not the state of art, it is very simple. That is why we use Algorithm 2 in this experiment to evaluate the performance of the structure learning after identifying candidate parent sets.
>
> The networks of this size is also not the state of art. The 6 networks in our experiment can be optimized exactly. Therefore, the experimental results with the 6 networks do not show the superiority of our heuristic method directly. However, from the comparison to the space of orderings [Teyssier and Koller, 2005] [Scanagatta et al, 2015], our experiment provide the indirect evidence to improve the performance of structure learning with larger networks. Due to our budget constraints, we targeted only 6 small networks that can be encoded within the 8192 bits of the second generation Fujitsu digital annealer. However, the third generation of Fujitsu digital annealer has the 100,000 bit capacity. Moreover, Fujitsu announced that they achieved a megabit-class performance with digital annealer. If a megabit-class digital annealer is available to us, we will be able to encode networks with a thousand variables into the circuit of digital annealer. Note that the ordering-based search and acyclic selection ordering-based search are the original version without tabu list. The number of restart  is the maximum within 6000 [s] plus the running time of Algorithm 1.
>
> We look forward to hearing from you regarding our submission.
> We would be glad to respond to any further questions and comments that you may have.

---

> > ### Comment · Reviewer_9dre · 2021-08-19
> > **Re: Author Response**
> >
> > I read the authors' response to my comments, as well as the other reviews and responses.
> >
> > I didn't repeat the experiments from Reviewer Yh3Z, but I believe they clearly show my concerns with this work. As early work in QUBO formulations, I believe this is interesting and would be a great fit for an appropriate workshop. As presented in the paper, the theoretical/encoding contributions are limited (though I do look forward to those hinted at in the response), and it seems BNSL may not be the best problem, in terms of low-hanging fruit, to really highlight the empirical advantages of digital annealers.
> >
> > Overall, I believe the work is of interest to a more focused audience, and I would suggest to look for such an audience. The hardware clearly matters here, and the 100kbit annealer may make things different in the future.

---

### Official Review · Reviewer_UDX7 · 2021-08-01

**Rating:** 6
**Confidence:** 1

**Summary:**

The premise of the paper is both novel and interesting, in regarding Bayes network structure learning as a combinatoric optimization problem amenable to simulated annealing, specifically with current quantum computing tools.  The paper offers an approach to converting a score-based Bayesian network structure learning into quadratic unconstrained binary optimization. The improvement in representation of Bayes networks structure over current published methods turns on identification of candidate parent sets, and that this shows improvements compared to "ordering space search algorithms"

**Ethical Concerns:**

There are no manifest ethical concerns with the work.

**Limitations And Societal Impact:**

The authors adequately address speculations about how quantum computing may have extensive social ramifications as it becomes applicable.

**Main Review:**

The annealing processor optimizes a quadratic function of the vector \sigma (Equation 3).It may be familiar to those working in the field on how this relates to a solution off a via simulated annealing, and how the network structure translates such as encoding network paths by use of bits available to the annealer.  Assuming the authors are addressing an audience that understands BN structure learning, an extended version of section 2.3 for those not familiar with quantum annealing would make this work accessible.   As it is, I am not in a position to offer an opinion on the method proposed, and give the paper the benefit of the doubt.

Given the complexity of the encoding, reliance on identification of parent sets, and how the method relies crucially on the bit capacity of current digital annealers, one gets the presumption that the progress that can be made in generally applicable improvements in structure learning is limited.


**Time Spent Reviewing:**

2

---

> ### Author Response · Authors · 2021-08-09
> **Author Response**
>
> We appreciate your kind comment.
>
> Due to our budget constraints, we targeted the networks with 27 to 70 variables that can be encoded within the 8192 bits of the second generation Fujitsu digital annealer. However, the third generation of Fujitsu digital annealer has the 100,000 bit capacity. Moreover, Fujitsu announced that they achieved a megabit-class performance with digital annealer. If a megabit-class digital annealer is available to us, we will be able to encode networks with a thousand variables into the circuit of digital annealer.
>
> Though the encoding process is complex, it does not depend on the problems. Therefore, we consider that there is no impediment to practical application due to the complexity. The performance of candidate parent set identification is known [de Campos et al, 2011]. We consider that it is necessary to analysis the dependance of Algorithm 1 on candidate parent sets.
>
> We agree with extending the section 2.3. We will make the clear connection between the section 2.3 and the section 4. Moreover, we will rearrange Figure 1 to represent the relationship between the digital annealer architecture and our conversion method.
>
> We look forward to hearing from you regarding our submission.
> We would be glad to respond to any further questions and comments that you may have.

---

### Decision · Program_Chairs · 2021-09-27

**Decision:**

Reject

**Comment:**

This work proposes some quite interesting ideas for learning the structure of Bayesian nets (BNSL). This has been acknowledge during the discussions. However, the current data/experiments/discussions do not show enough evidence that this is an effective method for BNSL (ideas show promise, but it was not considered enough yet). Reviewers and meta-reviewer think that ideas will be eventually worth publishing, so while this is a negative recommendation (sorry!), I would like to say a positive work: keep the interesting work going and you will find a suitable venue to discuss the work.